# Pathological Characteristics of Domestic Pigs Orally Infected with the Virus Strain Causing the First Reported African Swine Fever Outbreaks in Vietnam

**DOI:** 10.3390/pathogens12030393

**Published:** 2023-03-01

**Authors:** Thi Thu Huyen Nguyen, Van Tam Nguyen, Phuong Nam Le, Nguyen Tuan Anh Mai, Van Hieu Dong, Tran Anh Dao Bui, Thi Lan Nguyen, Aruna Ambagala, Van Phan Le

**Affiliations:** 1College of Veterinary Medicine, Vietnam National University of Agriculture, Hanoi 100000, Vietnam; 2Faculty of Animal Science and Veterinary Medicine, Bac Giang Agriculture and Forestry University, Bac Giang 230000, Vietnam; 3Institute of Veterinary Science and Technology (IVST), Hanoi 100000, Vietnam; 4National Centre for Foreign Animal Disease, Canadian Food Inspection Agency, Winnipeg, MB R3E 3M4, Canada

**Keywords:** ASFV, clinical signs, gross findings, orally experimental infection, pathology

## Abstract

African swine fever (ASF) is currently Vietnam’s most economically significant swine disease. The first ASF outbreak in Vietnam was reported in February 2019. In this study, VNUA/HY/ASF1 strain isolated from the first ASF outbreak was used to infect 10 eight-week-old pigs orally with 10^3^ HAD_50_ per animal. The pigs were observed daily for clinical signs, and whole blood samples were collected from each animal for viremia detection. Dead pigs were subjected to full post-mortem analyses. All 10 pigs displayed acute or subacute clinical signs and succumbed to the infection between 10 to 27 (19.8 ± 4.66) days post-inoculation (dpi). The onset of clinical signs started around 4–14 dpi. Viremia was observed in pigs from 6–16 dpi (11.2 ± 3.55). Enlarged, hyperemic, and hemorrhagic lymph nodes, enlarged spleen, pneumonia, and hydropericardium were observed at post-mortem examinations.

## 1. Introduction

African swine fever (ASF) is a viral disease that is highly contagious and devastating to domestic and wild pig populations. The economic impact of this disease is profound, with significant losses incurred globally due to high mortality rates and trade restrictions. Although ASF is primarily present in African countries, the disease has now spread to other areas such as Asia, Europe, and America [1]. ASF transmission occurs through direct contact between infected and susceptible pigs, but contaminated feed, water, equipment, and clothing can also spread the disease [2]. Once ASF enters a pig population, it spreads rapidly, leading to high fever, loss of appetite, and internal bleeding. Mortality rates can be as high as 100%, and currently, there are no specific treatments or vaccines for ASF [3]. Strict biosecurity measures and culling infected pigs are the only viable means of controlling the spread of the disease [4].

ASF virus (ASFV), the causative agent of African swine fever, is a large, enveloped virus with a double-stranded DNA genome. It belongs to the Asfivirus genus within the *Asfarviridae* family. The viral genome of ASFV ranges from 170 to 190 kbp in length, encoding over 170 proteins [5,6]. ASFV strains are divided into 24 genotypes based on the partial sequence of the ASFV B646L gene encoding p72 protein [7]. ASFV can be further sub-genotyped through various methods, which target different viral genome regions to identify genetic variations among different virus strains. Common sub-genotyping methods for ASFV include the analysis of specific genes, such as the E183L gene encoding p54 protein (genotype I-XXIV), EP402R gene encoding CD2v protein (serotype I-VIII), and B602L gene (multiple variants), as well as the intergenic region (IGR) between MGF 505-9R and MGF 505-10R, and other genes like K205R, MGF 110-14L, H240R, O174L, E199L, K145R, I9R, and MGF 505-5R [8,9,10,11,12,13,14,15,16]. Sub-genotyping ASFV can help researchers better understand the virus’s epidemiology and evolution, which is essential for developing effective control and prevention measures against this economically important disease.

ASF was first reported in Kenya in 1921 [17]. The first ASF transcontinental spread was reported in 1957 from Africa to Spain. The virus, which belonged to ASFV p72 genotype I, then spread to Portugal (1960) and the neighboring countries. ASF was eliminated from Europe in the mid-1990s, except in Sardinia, which remains an endemic [18,19]. The second ASFV transcontinental spread happened in 2007 [8,18,19]. An ASFV p72 genotype II was detected in Georgia and later spread to Armenia, the Russian Federation, Haiti regions, and eastern European countries mainly through wild boars [19,20].

In Asia, the first ASF outbreak was reported in mainland China in August 2018. ASFV then rapidly spread across China to almost all the neighboring countries, including Vietnam, Mongolia, Cambodia, Laos, Myanmar, Hong Kong, North Korea, South Korea, the Philippines, India, Indonesia, and Thailand. The responsible virus was a highly virulent ASFV strain belonging to ASFV p72 genotype II [21]. In 2021, besides genotype II, genotype I ASFVs were isolated from lymph nodes and spleens in China, which caused chronic forms including intermittent fever, arthroncus, and cutaneous necrosis [22]. 

Previous studies showed that ASF could manifest in four clinical forms, depending on the virus strain, transmission route, dose, and the host’s characteristics. These four clinical forms are peracute, acute, subacute, and chronic. In peracute cases, pigs may die suddenly without showing any prior clinical signs. In acute cases, pigs may exhibit high fever, anorexia, lethargy, and hemorrhages on the skin and internal organs. In subacute and chronic cases, pigs may show milder clinical signs, such as loss of appetite, weight loss, and fever [23]. ASFV strains are classified based on their virulence, which can significantly impact the severity of the disease. Highly virulent strains can cause mortality rates of up to 100%, while low virulent strains may cause only mild or asymptomatic infections. Once infected, pigs shed the virus at high levels in all secretions and excretions, contributing to the rapid spread of the disease. Effective control measures, including strict biosecurity protocols and rapid detection, are essential to prevent the spread of ASFV and limit its impact on the swine industry [1].

In Vietnam, the first two outbreaks of ASF were reported in two adjacent northern provinces, Hung Yen and Thai Binh, in the red river delta in early 2019 [14,21]. After the first two outbreaks, the virus quickly spread to all 63 provinces in Vietnam. The highly contagious nature of ASF and the lack of an effective vaccine or treatment for the disease led to the mass culling of infected and contact pigs as a control measure. The impact of ASF on the pig industry in Vietnam has been significant, with nearly 5.9 million pigs destroyed, accounting for 20% of the total swine population in the country. This has led to a pork shortage and a sharp increase in the prices of pork products. Many small-scale pig farmers, who make up the majority of the pig industry in Vietnam, have been severely affected by the outbreak, with some losing their entire herds and livelihoods (FAO & Ministry of Agriculture and Rural Development, Vietnam). The results of the genetic sequencing analysis indicated that the ASFV strains responsible for the initial outbreaks in Vietnam belonged to genotype II, which is comparable to the ASF Georgia 2007/1 strain [21]. Although there have been many studies on the molecular epidemiology of ASFV strains causing disease in pigs in Vietnam, studies on their pathology are still very limited [24]. A better understanding of the virulence of ASFV strains is crucial for effectively controlling and managing ASF. Therefore this study was conducted to understand better the pathogenicity of the ASFV strain that caused the first reported outbreak in Vietnam. 

## 2. Materials and Methods

### 2.1. Ethics Statements

The animal experiment was conducted at the large animal biosafety level 2 facility according to the animal use guidelines at the Vietnam National University of Agriculture (VNUA), Hanoi, Vietnam.

### 2.2. Virus Strain 

The virus strain used for the study was VNUA/HY/ASF1 (GenBank Accession no. MK554698), which belonged to p72 genotype II and originated from infected pigs during the first outbreak of ASF in Vietnam in February 2019 [21]. Healthy pigs aged 8 to 10 weeks were used to collect primary porcine alveolar macrophages (PAMs) for ASFV culture. Real-time PCR was performed to confirm the absence of various viruses such as Porcine circovirus type 2 (PCV2), Classical swine fever (CSF), Porcine reproductive and respiratory syndrome (PRRS), and African swine fever (ASF) using a kit from Median Diagnostics Inc. (http://www.mediandiagnostics.com, accessed on 19 May 2021). The PAM cells were cultured in RPMI 1640 medium (Gibco) supplemented with 10% fetal bovine serum (FBS) and 1% antibiotic. The cells were seeded at a density of approximately 4 × 10^5^ cells/cm^2^ onto tissue culture plastic plates. After 48 h of ASFV infection, 20 µL of 1% porcine red blood cells in RPMI medium was added to each well of PAM cells. The formation of hemadsorption (HAD) rosettes on ASFV-infected PAM cells was observed daily under an inverted microscope for five days. As previously described, the virus titer was determined using the Hemadsorption (HAD) assay and calculated using the Reed and Muench method [25,26].

### 2.3. Animals 

A total of 15 healthy eight-week-old crossbred (Yorkshire - Landrace) pigs were used in this study. All pigs were previously confirmed to be negative for African swine fever virus, porcine circovirus 2, foot-and-mouth virus; classical swine fever virus, and porcine reproductive and respiratory syndrome virus by conventional PCR and real-time PCR (data not shown). Pigs were also tested negative for antibodies to ASFV by enzyme-linked immunosorbent assay (VDPro^®^ ASFV Ab i-ELISA ver 2.0 Kit, Median Diagnostics, Seoul, Republic of Korea). Pigs were housed at the biosafety level 2 large animal facilities at the Vietnam National University of Agriculture, Hanoi, Vietnam, provided ad libitum feed and water, and observed daily. After one week of acclimatization, pigs were randomly divided into two groups, one with 10 pigs (ASFV-infected group) and the other with 5 pigs (mock-infected group), and moved to two separate pens. Five pigs in the control group were inoculated with one ml of sterile DMEM. Pigs were monitored for clinical signs, and their rectal temperatures were measured using a digital thermometer daily.

### 2.4. Sample Collection

In this study, blood and oral fluid samples were collected from each pig to assess the ASFV load using a kit from Median Diagnostics Inc. (http://www.mediandiagnostics.com, accessed on 19 May 2021). The cotton rope chewing method was used to collect oral fluid samples, which had previously been used in a similar study [27]. In detail, the pigs were allowed to chew on the rope for 45 min until the rope was sufficiently wet. The wet rope was then compressed into plastic bags to recover the oral fluid. Approximately 2.5 mL of the oral fluid was transferred into a 15 mL tube for analysis. Blood samples were collected from the jugular vein of each pig before feeding in the morning at 0, 2, 4, 6, 8, 10, and 12 dpi or daily when the trial pigs showed clinical signs (e.g., fever) for viremia detection and not causing stress on experimental pigs. The blood samples were then placed into blood collection tubes containing an anticoagulant solution (EDTA). After the pigs died, a necropsy was conducted immediately by designated veterinarians, and organ samples were collected for ASFV genome detection. The results of the DNA load analysis of both oral fluid and blood samples and the organ samples were used to assess the severity of ASFV infection in the pigs.

### 2.5. DNA Extraction and Real-Time PCR 

To detect ASFV, DNA was extracted from the whole blood, oral fluid, and tissue samples obtained from necropsied pigs. The DNeasy Blood & Tissue Kit (Qiagen, Hilden, Germany) was used to extract DNA from whole blood and oral fluid samples, while DNA was extracted from each homogenized tissue sample using the DNA Mini Kit (Qiagen, Hilden, Germany) following the manufacturer’s instructions. ASFV was detected using the VDx ASFV qPCR kit (Median Diagnostics). Briefly, 5 μL of extracted DNA was mixed with 10 μL of 2× master mix and 5 μL of 4× oligo mix in a PCR tube. The reaction proceeded under the following conditions: 40 cycles of 95 °C for 15 s and 58 °C for 60 s using a CFX96 Touch Real-Time PCR Detection System (Bio-Rad Laboratories Ltd., Hercules, CA, USA). Samples with a Ct (cycle threshold) value less than 40 were considered positive for ASFV.

### 2.6. Scoring of ASF Symptoms and Gross Pathology Findings

To evaluate the virulence of the ASFV strain VNUA/HY/ASF1, the clinical scoring system was used as described previously [24,28]. For each criterion, a score was recorded for either normal (score 0), slightly altered (score 1), distinct clinical symptom (score 2), or severe ASF symptom (score 3). The scores of all parameters and individual pigs’ total scores were estimated daily. The possible total score was a maximum of 18. Pigs with a total clinical score higher than 3 were defined as pigs having ASFV infection. Any dead pigs were immediately necropsied and through post-mortem. When any pigs had severe clinical signs (fever, anorexia, cough, diarrhea, and so on) for more than two consecutive days, or when the total accumulated clinical sign score exceeded 18 points, which was the humane endpoint for the diseased pigs. These diseased pigs were treated with anesthesia drugs and disposed of humanely [20].

### 2.7. Statistical Analysis

Student-*t* test was used to evaluate differences between the two groups for statistical significance. Statistical analyses and data visualization were performed with SPSS statistic 20.

## 3. Results

### 3.1. Clinical Signs

Although pigs were inoculated with the same dose at the same time using the same route, the onset of clinical signs ranged from 4 to 14 dpi. Fever and lethargy were the first clinical signs observed. Two out of ten ASFV-infected pigs showed increasing rectal temperatures at 4 dpi. The remaining animals started fever later, and most showed intermittent fever (Table 1). None of the animals in the uninfected group developed fever (Figure 1). At 15 dpi, the mean rectal temperature of the pen increased to 40.8 °C (*p* < 0.05). Although the mean temperature during the first 3 days after virus infection differed between the ASFV-infected and un-infected pig groups, this difference was not statistically significant (*p* > 0.05).

Other clinical signs observed in the ASFV-infected group included lethargy, anorexia, fever (10/10), recumbency (8/10), cough (5/10 pigs), diarrhea (3/10 pigs), and skin hemorrhages (4/10 pigs). Lethargy was first observed in three out of ten pigs at 5–8 dpi (Table 1). The pigs succumbed to the infection between 10 to 27 (19.8 ± 4.66) dpi (Table 1 and Figure 2). In addition, the mean clinical scores of the ASFV-infected group changed over time. There was a statistically significant difference between the mean clinical scores of the ASFV-infected and uninfected groups (*p* > 0.05) from 6 to 23 dpi. The pigs of the uninfected group did not show clinical signs (Figure 3). Fever and lethargy were observed in all animals at least 4–6 days before the pigs succumbed to the infection. Skin hemorrhages, diarrhea, and cough were observed in a few animals (Figure 4). Despite the clinical sign majority of the animals stayed active with normal appetite until late in the infection, and two of the 10 infected animals stayed active until they were found dead the next day. Overall clinical sign findings in this study suggested acute and/or subacute ASF infection, as described in a previous report [23,29].

### 3.2. Blood and Oral Fluid Sample Analysis

Following oral inoculation, the first pig (#5) showed viremia at 6 dpi (Table 1). ASFV genome was detected in oral fluid starting at 9 dpi, three days after the first detection of viremia in pen (Figure 5). Despite mild fever and lethargy observed by 9 dpi, all animals remained active and had a normal appetite. All the pigs in the control group stayed healthy throughout the experiment and were negative for the ASFV genome by real-time PCR (data not shown).

### 3.3. Gross Pathological Findings

The spleen was significantly larger and dark red with congestion and hemorrhage (Figure 6A, red arrow) in all pigs who succumbed to the infection. Tonsil, mandibular, mediastinal, and inguinal lymph nodes were enlarged, hemorrhagic, and/or congested (Figure 6G, E). Mesenteric lymph nodes were also markedly red and enlarged (Figure 6A-red circle). The liver was enlarged, congested, and had scattered pale foci (Figure 6A, blue arrow). Petechial hemorrhages were observed on the cortex (Figure 6B, black arrow). Lungs were congested, hemorrhagic, and edematous, with moderate to severe interstitial pneumonia (Figure 6C). Kidneys were congested (Figure 6F). Hydropericardium, hemorrhages in gastric mucosa, and gallbladder wall were observed in some pigs (Figure 6D-black arrow, H and I-black arrow).

None of the pigs from the uninfected group showed ASF-related lesions (Figures not shown). 

## 4. Discussion

African swine fever (ASF) is a contagious hemorrhagic fever of domestic and wild swine. ASF has become a global veterinary concern since it was first reported in China in 2018. Later it rapidly spread to almost all Asian countries near China, including the most recent outbreaks in India [4]. Until now, there is no safe and effective ASF vaccine available worldwide. In Vietnam, the first ASFV outbreak was reported in Hung Yen province in February 2019. The infected pigs showed typical clinical signs of ASF, such as fever, lack of appetite, skin hemorrhages, bloody diarrhea, cyanosis, and 100% mortality. However, the clinical signs reported with the ASF outbreaks in Vietnam did not always align with the peracute or acute forms of the disease. The responsible VNUA/HY/ASF1 virus belonged to ASFV p72 genotype II [21]. During the second ASF-reported outbreak in Thai Binh province, pigs quickly showed anorexia and high fever for three days before they were found dead [30]. The ASFV VNUA/HY/ASF1 isolate had been inoculated intramuscularly (titer, 10^3.5^ HAD_50_/mL per pig) into ten 7-to-8-week-old pigs (Yorkshire × Landrace × Duroc), and all of the inoculated pigs died within 5–8 dpi. Fever developed in three pigs at 3 dpi, and viremia was noted in those pigs at around 2 dpi. Oral fluid collected from the pigs tested positive for the ASFV genome as early as 3 dpi and continued to be positive until the end of the study [24]. Previous studies showed that the clinical outcome of ASF depends on many factors, including the pathogenicity of the virus isolates, the dose, the route of infection, and host characteristics [23,31]. Unlike CSF, which mainly affects young pigs, all age groups are considered equally susceptible to ASF. For easy handling, pigs 8 to 12 weeks old are often chosen to evaluate the pathogenicity of the ASFV strains by intramuscular, intranasal inoculations, and direct contact [32,33,34]. In addition, the oral inoculation dose of 10^3^ HAD_50_/mL per pig has been used in several studies to assess clinical signs of ASFV-infected pigs [35,36]. Therefore, in this study, VNUA/HY/ASF1 strain isolated from the first ASF outbreak was used to infect 10 eight-week-old pigs orally with 10^3^ HAD_50_ per animal.

In this study, compared to the previous study that used intramuscular inoculation, oral inoculation of the ASFV VNUA/HY/ASF1 strain resulted in a delayed onset and protracted clinical disease in pigs. Two of the orally inoculated pigs developed a mild fever at 4 dpi; viremia was noted only at 6 dpi in one of the pigs. The first animal dies of ASF at 10 dpi, and the last one at 27 dpi. ASFV genome was detected in oral fluid only after 9 dpi. Out of 10 pigs inoculated orally, only four got infected within 8 days post-inoculation. This suggests that the remaining six animals acquired the infection through direct contact with the four pigs that got infected following experimental inoculation. In a recent study conducted by Niederwerder et al. [35], the infection probability of ASFV Georgia 2007 via oral route following a single dose of 10^3^ TCID_50_ was calculated to be around 83.3%. Surprisingly our study only showed a 40% infection rate, which could be due to many reasons, including the method of inoculation, breed, and health of the animals, etc. Despite these changes, clinical signs observed in this experiment were similar to those reported before (fever, anorexia, depression, diarrhea) [24,30]. The appearance of clinical symptoms such as fever and lethargy was expected to appear after or coincide with the induction of viremia [24,37,38]. However, several pigs in this study developed fever and lethargy 2–4 days before the viremia was observed. This could be due to many reasons, including concurrent infections, ambient temperature fluctuation in animal pens, and handling of pigs for blood collection. The pigs used in the experiment were purchased from a commercial farm in Vietnam. They were not specific-pathogen-free animals. Therefore, it is possible that some of the pigs in the study were infected with common bacterial and viral agents found in swine herds in Vietnam. Such infections can induce fever and lethargy irrespective of the ASFV infection [39]. Hence, some non-ASF-specific clinical signs, such as fever and lethargy, could appear early or at the same time as ASF viremia in some animals. Therefore, we believe that the clinical picture observed in this study is closer to what happens in Vietnam and is more aligned with acute and subacute forms of the disease. No pigs died suddenly (per acute), and the mean time to death of pigs after oral inoculation was 19.8 ± 4.66 days. The first animal died of ASF infection at 10 dpi, and almost all animals stayed healthy despite an ongoing ASF infection in the pen until 20 dpi when 60% of the animals in the pen died. Despite protracted pathogenicity, all pigs that died of ASF had similar gross pathological lesions, i.e., splenomegaly and hemorrhagic lymphadenitis. Interestingly, in five animals, we also observed meningeal hemorrhages. In this study, there was a fluctuation of the Ct value obtained from oral fluid samples of pigs during the infection process, which could be affected by many factors, such as the health conditions after ASFV infection and chewing activities on the ropes. It was noted that some ASFV-infected pigs could not chew on ropes when they had severe clinical symptoms. The variations of the Ct value obtained from the oral fluid were also reported by Goonewardene, K. B. et al. [40].

As observed in this study, the protracted clinical picture following ASF’s introduction into the pig pens has significant practical relevance. As the observations from the study point out, ASF infections can quickly spread unnoticed and/or slowly on the farm for two to three weeks once the farm is infected with ASFV. Therefore, any abnormal behavior in a pig farm should be quickly addressed by submitting samples for ASF detection. Such an approach to contain and control ASF spread, especially in ASF-endemic countries such as Vietnam.

## 5. Conclusions

In this study, we evaluated the pathogenicity of the ASFV VNUA/HY/ASF1 strain following oral inoculation of 8 weeks old commercial pigs. Pigs infected showed acute to subacute clinical form with a mean time of death around 19.8 ± 4.66 days. The onset of the clinical signs of different pigs was also different, ranging from 4 to 14 dpi. The viremia pigs infected with ASFV were detected only after 6 dpi (11.2 ± 3.55). This study provides valuable information regarding the pathogenesis of ASF outbreak strain VNUA/HY/ASF1 in Vietnam following oral infection in domestic pigs—a similar model to the natural exposure of ASF in the field.

## Figures and Tables

**Figure 1 pathogens-12-00393-f001:**
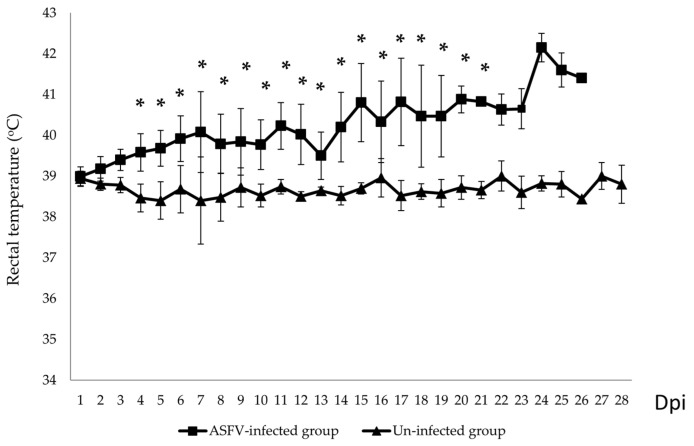
Mean of daily rectal temperatures (°C) of the ASFV- infected and un-infected pig groups. * *p* < 0.05.

**Figure 2 pathogens-12-00393-f002:**
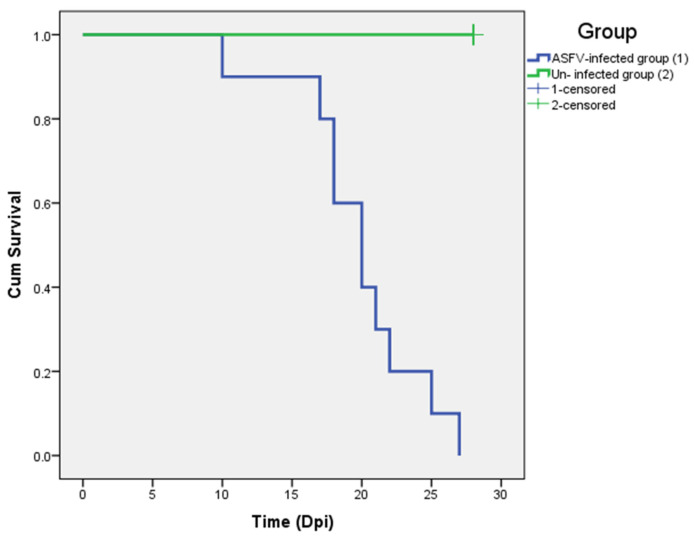
The survival rate of ASFV-infected (blue) and un-infected (green) pigs.

**Figure 3 pathogens-12-00393-f003:**
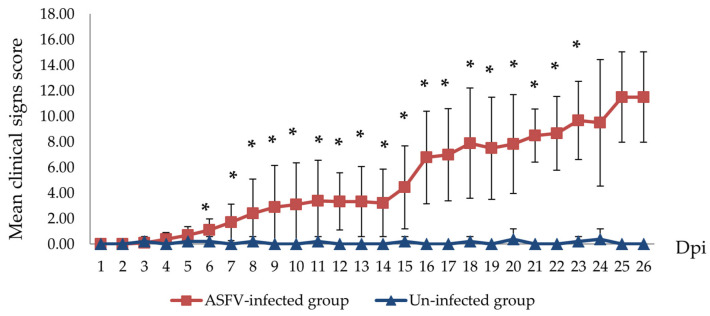
Mean clinical scores of ASFV-infected (red line) and un-infected (blue line) pig groups. The scores were calculated as previously described by I. Galindo-Cardiel and H. S. Lee [24,28]. (* *p* < 0.05).

**Figure 4 pathogens-12-00393-f004:**
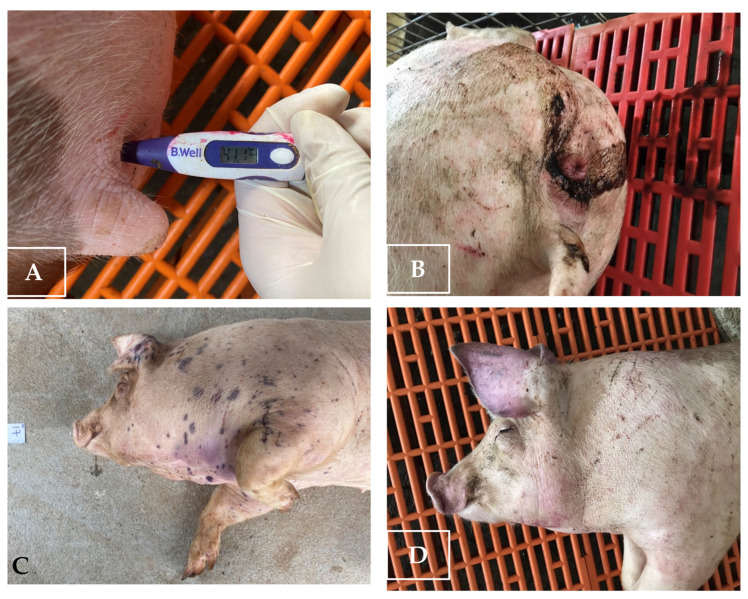
Some of the clinical signs of pigs infected with ASFV. (**A**) Fever; (**B**) diarrhea; (**C**) petechial hemorrhages on the skin; (**D**) erythema and skin hemorrhages.

**Figure 5 pathogens-12-00393-f005:**
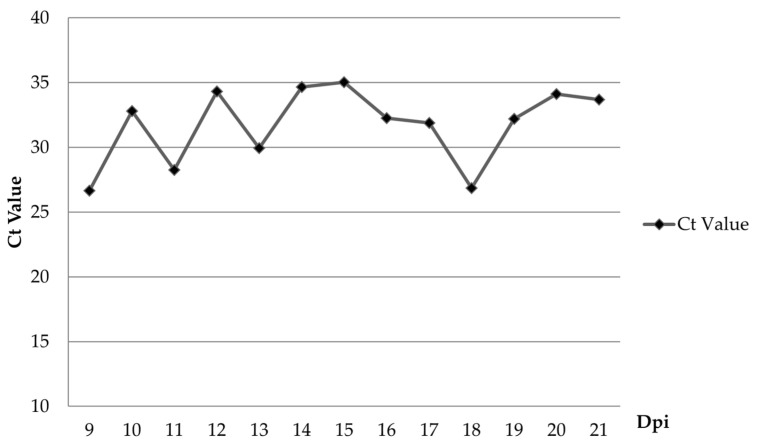
The mean viral load (Ct value) of the ASFV in oral fluids collected from the ASFV-infected pig group.

**Figure 6 pathogens-12-00393-f006:**
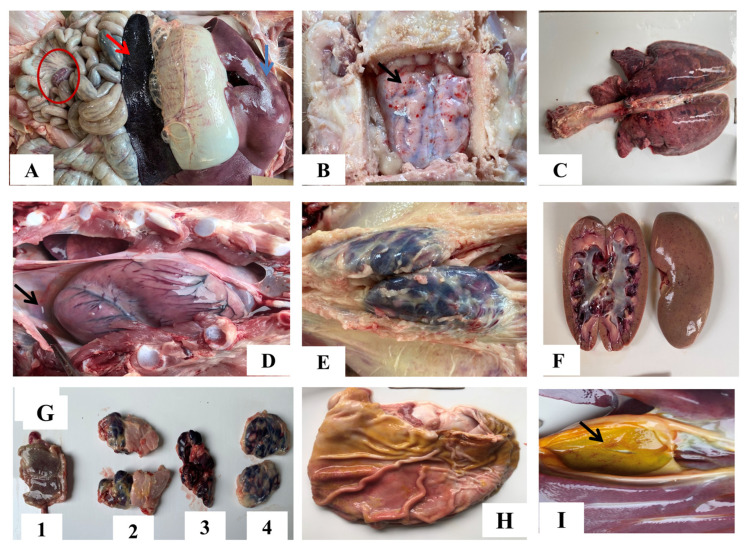
Gross pathological lesions were observed in pigs infected with ASFV. (**A**) Congested and enlarged spleen (red arrow); hemorrhagic mesenteric lymph nodes (red circle); enlarged liver with hemorrhages (blue arrow); (**B**) hemorrhagic meninges; (**C**) congested and interstitial pneumonia; (**D**) hydropericardium (black arrow); (**E**) enlarged and hemorrhagic inguinal lymph nodes; (**F**) petechial hemorrhage in kidneys; (**G1**) hemorrhagic tonsils; (**G2**) enlarged and hemorrhagic mandibular lymph nodes; (**G3**) mediastinal lymph nodes; and (**G4**) inguinal lymph nodes; (**H**) gastric mucosa with superficial lesions and hemorrhage; (**I**) petechiae gallbladder wall (black arrow).

**Table 1 pathogens-12-00393-t001:** Dates (shown as dpi) of onset of clinical signs and viremia.

No.	Date of Onset of Clinical Manifestations							Dead	Onset of Viremia
	Anorexia	Recumbency	Diarrhea	Cough	Lethargy	Fever	Skin Hemorrhages		
1	14	15	14	11	5	4	15	18	8
2	14	16	-	-	11	11	-	21	12
3	16	19	18	11	10	15	-	20	14
4	22	23	-	-	12	19	-	25	16
5	9	-	-	5	6	4	8	10	6
6	18	19	-	-	14	15	-	22	12
7	22	23	-	-	12	20	26	27	16
8	15	-	-	7	8	8	-	18	8
9	16	19	19	-	12	15	-	20	8
10	14	15	-	15	12	11	15	17	12

## Data Availability

The data presented in this study are available on request from the corresponding author.

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
