# Peer review of "Pathological Characteristics of Domestic Pigs Orally Infected with the Virus Strain Causing the First Reported African Swine Fever Outbreaks in Vietnam"

_pathogens, 2023, doi:10.3390/pathogens12030393_

Round 1

Reviewer 1 Report

The manuscript submitted by Nguyen et al. entitled "Pathological Characteristics of Domestic Pigs Orally Infected with the Virus Strain Causing the First Reported African Swine Fever Outbreaks in Vietnam" to the SI "Emergence and Control of African Swine Fever" aims to compare the oral inoculation of the ASFV VNUA/NY/ASF1 strain to a previous study that used intramuscular inoculation. The main conclusion of this manuscript is that occurs a delayed in the symptoms. Pigs inoculated showed acute to subacute clinical forms of ASF with a mean time of death around 20 days, with a onset of the clinical signs ranging from 4 to 14 dpi. Interestingly, the viremia was only detected after the 6 dpi, sometimes at day 14 pi. In the opinion of this reviewer, this study provides important information regarding the pathogenesis of the ASFV VNUA/NY/ASF1 strain, involved in outbreak in Vietnam.

The manuscript is well-written, with a M&M section clear and Results were clear. However several points needs to be improved before the acceptance of the manuscript for publication:

- In the Introduction, authors should emphasize the absence of antiviral and vaccines for ASFV. The also add a more recent review on this theme (https://doi.org/10.1080/22221751.2022.2108342)

- In the discution, the authors should discuss the effect of the animals' age on the immune responses against ASFV. It is not clear why young pigs were used in this study (8 weeks).

- Authors  should also explain the selected amount of virus to be inoculated in pigs 10^3 HAD50.

- Finally, authors should better discuss with symptoms appear before viraemia in the inoculated animals.

Reviewer 2 Report

1. Only one dose (1000 HAD50) ASFV was chosen for an oral challenge experiment in domestic pigs, and the work is only a part of a paper published in 2013 by Howey EB et al in journal Virus Research. The dose is not in agreement with that in any certain natural clinical infection, so a detailed discussion is needed.

2. The animal experiment was conducted at a biosafety level 2 facility. This is not allowed acoording to OIE guidelines and so no humane endpoints were stipulated and therefore this is not an experiment under good animal wellfare and ethics.

3. The virus strain used is a virulent ASFV, and it causes 100% death in the inoculated pigs. So the conclusion that the virus caused a subacute infection is not appropriate and not correct.

4. Figure 1 is confusing. The rectal temperature within the first 3 days is diverse between the infected and the uninfected control pigs, how to explain?

5. Figure 5 needs explanation too, expecially the variations of the Ct value during the infection process.

The authors set up a scoring standard, so  they should indicate th humane endpoits for the diseased pigs.

Reviewer 3 Report

Dear editor, 

The manuscript presented by Thi Thu Huyen Nguyen et al., is another example of biological properties of ASFV that has been circulating since 2007. 

The work seems good, methods and results are described well, and the conclusions made by the authors are clear and related to the results. 

Of course I have some comments, that I added in the .pdf file uploaded here, I hope my comments will be useful and answered well. 
Also, I would recommend professional English editing. 

Round 2

Reviewer 1 Report

The authors answered to all questions raised by this reviewer. Thus, the manuscript is suitable for publication at the present form.